# Effect of Technology Acceptance on Blended Learning Satisfaction: The Serial Mediation of Emotional Experience, Social Belonging, and Higher-Order Thinking

**DOI:** 10.3390/ijerph20054442

**Published:** 2023-03-02

**Authors:** Tianjiao Chen, Heng Luo, Qinna Feng, Gege Li

**Affiliations:** Faculty of Artificial Intelligence in Education, Central China Normal University, Wuhan 430079, China

**Keywords:** blended learning, emotional experience, higher-order thinking, learning satisfaction, social belonging, technology acceptance

## Abstract

This study explored the relationship between technology acceptance and learning satisfaction in the context of blended learning, with a particular focus on the mediating effects of online behaviors, emotional experience, social belonging, and higher-order thinking. A total of 110 Chinese university students participated in this study and completed a questionnaire at the end of 11 weeks of blended learning. The results demonstrate that technology acceptance directly and indirectly relates to blended learning satisfaction. The mediation analysis further revealed two significant mediating pathways from technology acceptance to blended learning satisfaction: one through higher-order thinking, and the other through serial mediation of emotional experience, social belonging, and higher-order thinking. Moreover, there was no significant mediating effect of online learning behaviors on blended learning satisfaction. Based on these results, we have proposed practical implications for improving blended learning practice to promote learner satisfaction. These results contribute to our understanding of blended learning as an integrated construct under the triadic interplay of technical environment, learning behaviors, and individual perceptions.

## 1. Introduction

Blended learning has been an important feature of higher education practice for the past few decades [1,2,3], and has emerged as a mainstream instructional model in the post-COVID-19 era [4]. Its proven effectiveness in providing flexible, timely, and continuous learning [5], has helped enhance learning engagement [6], and promote learning outcomes and experience [7,8,9]. By combining the advantages of face-to-face and online learning [10], this method emphasizes the central role of the online community created by the learning management system (LMS), which promotes student dialogue, reflection, and communication [11]. Much like a traditional classroom, the knowledge acquisition of blended learning usually involves behavioral, cognitive, emotional, and personal factors, and environmental events that all operate as interacting determinants, as suggested by social cognitive theories [12,13]. To extend our understanding of blended learning, for improved quality and efficacy, the perspective provided by social cognitive theory warrants particular attention.

Learning satisfaction is defined as students’ perception of their course learning experience and the perceived value of the education received when attending an educational activity [14,15]. It is important to note that the current prevalence of blended learning practice has been largely forced upon learners, due to COVID-19 related school closures. To ensure the long-term and sustainable implementation of blended learning in the post-pandemic era, learner satisfaction merits our special attention, as it is known to significantly predict the acceptance and completion of blended courses [16,17,18]. While previous studies have investigated learning satisfaction as an outcome variable, rather than predictor [19,20], studies that integrate crucial factors determining learning satisfaction with blended learning, such as behavior, individual emotion, cognition, and environment, are sorely lacking.

Triadic reciprocal determinism (TRD), proposed by Bandura, is often used as a conceptual model to represent a constant dynamic interaction among three elements: environment, behavior, and personal characteristics [21,22,23]. When students engage in blended learning, the environment refers to both the physical environment of the traditional classroom and the perceived online community [24]. The behaviors of online learning are often measured by login time and forum posts [25], and personal characteristics are those related concepts that can affect learning and behavior, including emotion, cognition, and motivation [23]. The correlational effects between environment and satisfaction have been previously reported, with a positive perception of the environment leading to improved satisfaction [16,26].

Although the TRD framework conceptually explains learning satisfaction, the inter-relationships between environment, behavior, and personal characteristics are rarely investigated, especially in the context of blended learning. Consequently, the influencing directions among key constructs and serial paths of influence need to be identified, and verified with empirical data. From the perspective of social cognitive theory, this study tests a new theoretical model based on the framework of TRD. The proposed model examines the relationship between students’ technological acceptance of online learning environments and their learning satisfaction, and takes individual characteristics and behaviors as a series of mediating variables. The present study aims to further our understanding of the interplay between the environment, the individual, behavior, and blended learning satisfaction, to provide a guide for the instructional practice of blended learning.

## 2. Literature Review

### 2.1. Blended Learning as a Technology-Enhanced Environment

A large and growing body of literature, demonstrates that the learning environment affects learning satisfaction though personal behavior, cognition, and emotional state [27,28]. Traditionally, learning environments have been defined in terms of the physical and social environments in a classroom setting [29], although Piccoli, Ahmad [30] expanded this definition by identifying differences between e-learning environments and classroom learning environments, including technology, content, interaction, learning model, and learner control. These factors can be classified into technological environment and social environment.

The technological environment of blended learning refers to the LMS built by information and communication technologies. Therefore, the technology acceptance model (TAM) is effective in explaining students’ satisfaction of blended learning as a technology-enhanced learning environment. First proposed by Davis [31], this model combines two of the most influential factors: perceived ease of use (PEU) and perceived usefulness (PU). While most studies have investigated these two constructs separately, we combined PU and PEU into technology acceptance, as a key construct of the blended learning environment. The rapid development of LMSs in recent years has reduced the variance in PU and PEU, and has led to a high correlation and possible multicollinearity between the two [32]; while several studies have revealed that technology acceptance significantly predicts students’ learning satisfaction in technology-enhanced learning environments [19,26,33,34]. Based on the above review results, we proposed the following hypothesis:

**H1(c’).** *Learners’ technology acceptance of the LMS is positively related to learning satisfaction*.

### 2.2. Login and Post Behaviors as Mediators of Blended Learning Satisfaction

Behavior is one of the core constructs in TRD, representing the individual’s willingness to actively learn, which is usually defined as observable actions and reactions [35]. When students participated in blended learning, their learning behaviors could be extracted from LMS logs data, which could be further classified as posting behaviors (e.g., number of forum posts, number of reply posts) and non-posting behaviors (e.g., total login time, total number of clicks) [25]. The prior literature suggests that posting behavior is a symbol of student interaction and opinion expression in online learning, and represents students’ active learning [36]. As for non-posting behavior, a review study noted that total login time was found to be a significant predictor of learning performance [25], as there are many lurkers who logged in to the discussion forum to speculate rather than to contribute. According to Amichai-Hamburger, Gazit [37], lurkers might still benefit from non-posting behaviors, such as browsing posts and active reflection.

The existing literature has reported the relationships between technology environment, behavior, and learning satisfaction, and suggests a mutual influence among the variables. Particularly, compared with the online and face-to-face learning contexts, blended learning reports a stronger relationship between learning behavior and satisfaction, since its multimodality and flexibility place higher demands on students’ self-regulated learning behaviors [38]. Most current research studies on technology and behavior, such as TAM and its extension models (e.g., TAM2, technology satisfaction model), have provided a theoretical framework to understand the relationship between technology and behavior [39,40], which all emphasizes that learners’ behaviors can be influenced by technology acceptance [31,41]. However, to date, there have been insufficient research studies investigating the influence of learner behavior on learning satisfaction, in the context of blended learning. Only a small body of literature indicated that behavior intention can positively predict satisfaction [42], but the influence of specific behaviors, such as login time and total number of posts, on satisfaction is not clear. Thus, to have a better understanding of the relationship between technology, learners’ behaviors, and learning satisfaction, we proposed the following hypotheses:

**H2(a1b1).** *Learners’ login time mediates the relationship between technology acceptance and learning satisfaction*.

**H3(a2b2).** *Learners’ posts mediate the relationship between technology acceptance and learning satisfaction*.

**H4(a1d21b2).** *Learners’ login time and posts serially mediate the relationship between technology acceptance and learning satisfaction*.

### 2.3. Emotional State and Cognitive Level as Mediators of Online Learning Satisfaction

The TRD has indicated bidirectional relationships between personal (emotional state and cognitive level), behavioral, and environmental factors [12]. Emotion encompasses the feelings, affection, and general moods that a learner brings to a task [35]. In blended learning, emotion refers to the feelings that students have toward the LMS, peers, and online learning activities [43], which can be conceptualized as an emotional experience acquired in the process of blended learning. Most studies found that students’ positive emotional experience of the LMS is influenced by technology acceptance [44]. For example, Padilla-Meléndez, del Aguila-Obra [45] reported that students are more likely to be interested in the learning process when they consider the platform useful, which further promotes improved learning satisfaction [46]. Furthermore, emotional experience also mediates the relationship between technology acceptance and satisfaction. For example, Gao, Jiang [47] conducted an experiment in the context of tourism, which has shown that the usefulness and ease of use of an LMS can indirectly affect students’ satisfaction, through emotional participation.

The second personal factor examined in this study was cognition level, which reflects students’ psychological engagement in learning activities [48]. According to Bloom’s taxonomy of cognitive objectives [49,50], students’ higher-order thinking activities, such as application and creation in blended learning, indicate a high level of cognitive engagement, which profoundly influences students’ learning satisfaction [51], particularly in the context of higher education, where college students with independent thinking have higher requirements for the development of their higher-order thinking skills [52].

Prior studies suggest a positive correlation between higher-order thinking and satisfaction. In addition, Gao, Jiang [47] demonstrated that the perceived usefulness of the learning platform has the greatest impact on students’ cognitive participation in blended learning. Furthermore, Manwaring, Larsen [6] revealed that students who have a positive sense that they can perform an academic task, and perceive that the task has value or interest to them, will experience enjoyment (a key component of emotional experience) and will be willing to exert more cognitive effort toward the activity. Hence, we investigated emotional experience and higher-order thinking as mediating variables between technology acceptance and learning satisfaction, and put forward the following hypotheses:

**H5(a3b3).** *Learners’ emotional experience mediates the relationship between technology acceptance and learning satisfaction*.

**H6(a5b5).** *Learners’ higher-order thinking mediates the relationship between technology acceptance and learning satisfaction*.

**H7(a3d53b5).** *Learners’ emotional experience and higher-order thinking serially mediate the relationship between technology acceptance and learning satisfaction*.

### 2.4. Social Belonging of Blended Learning: Antecedents and Consequences

In addition to TRD, the social cognitive theory (SCT) emphasizes the importance of social belonging, during learning processes. According to Bandura [12], a person’s behavior is partially influenced by social community, and strong community ties could provide important environmental conditions for knowledge exchange [53]. Whether students benefit from social influence is directly reflected by the acquisition of a sense of social belonging: a stronger sense of social belonging creates more intimate connections. Previous studies have demonstrated a positive relationship between social belonging and learning satisfaction [54], and online social interaction with other students is an important factor in motivating students to be satisfied with online learning [55]. In addition, due to the technical characteristics of the LMS, students are more likely to use the platform to post and discuss when they think the platform is both easy to use and useful. Therefore, higher technology acceptance leads to increased social belonging.

Social belonging plays a bridging role between emotional state and cognition level. First, social belonging provides opportunities for students to develop higher-order thinking: closer social ties promote the development of self-cognition [56], because social interaction among members of online communities, increases knowledge exchange. So and Brush [57] reported that group members share knowledge and skills through discussion, and establish a sense of connection, both of which support in-depth communication and meaningful dialogue among group members and promote the development of higher-order thinking. Students also gain social belonging in blended learning through emotional experience. Because of the lack of a prominent sense of intimacy and immediacy in online interaction [58], students’ pleasant emotional experience of LMS is conducive to their continuous participation in discussions. Delahunty, Verenikina [59] suggested that emotional state is necessary for individuals to construct their identities, respond to interactions in the learning community, and form a sense of belonging. Consequently, we propose the following hypotheses regarding social belonging variables that influence emotion and cognition:

**H8(a4b4).** *Learners’ social belonging mediates the relationship between technology acceptance and learning satisfaction*.

**H9(a3d43b4).** *Learners’ emotional experience and social belonging serially mediate the relationship between technology acceptance and learning satisfaction*.

**H10(a4d54b5).** *Learners’ social belonging and higher-order thinking serially mediate the relationship between technology acceptance and learning satisfaction*.

**H11(a3d43d54b5).** *Learners’ emotional experience, social belonging, and higher-order thinking serially mediate the relationship between technology acceptance and learning satisfaction*.

As part of social cognitive theory, the TRD provides us with a framework for the interrelation between the environment, behavior, and personal factors. In the context of blended learning, their interrelationships remain to be clarified, to further understand the serial process of how technology acceptance influences the learning satisfaction through emotional, cognitive, and behavior factors. Therefore, a conceptual model, composed of the hypotheses outlined above, was constructed, as shown in Figure 1.

## 3. Method

### 3.1. Ethics Statement

This study was conducted in accordance with the ethical standards of the Helsinki Declaration. The research procedures and instruments were reviewed and approved by the Institutional Review Board of Central China Normal University (CCNU-IRB-202103019, approved on 16 March 2021). Informed consent was obtained from all participants before conducting the study. All participants were made aware that their participation was voluntary, and their personal identifiable information would be kept anonymous from all publications and presentations. Participants could withdraw from the research study anytime, without penalty.

### 3.2. Research Design

In this study, a cross-sectional, nonexperimental approach for studying mediators was used to test the hypotheses. Cross-sectional, nonexperimental design enables identification of mediational effects from a statistical perspective, and treats mediators as clues of possible mechanisms of change [60], and thus is suitable for the present study, that seeks to explore the causal mechanism between technology acceptance and blended learning satisfaction through a series of mediating variables.

### 3.3. Instruments

A questionnaire was used to collect data, and included two major parts: a portion for the respondent’s basic data and another for responses to our research constructs (see Appendix A). The basic data portion recorded the subject’s demographic information (e.g., gender, age, and profession), while the second part recorded the subject’s perception of each variable in the model. This portion included items related to technology acceptance, emotional experience, social belonging, higher-order thinking, and learning satisfaction. All items were measured using a 5-point scale, ranging from 1 (strongly disagree) to 5 (strongly agree).

The technology acceptance questions were based on the questionnaire developed by Davis [31], which measures students’ technology acceptance of LMS, from perceived usefulness and perceived ease of use. Questions for emotional experience and social belonging were adopted from the questionnaire developed by [61], and captured students’ general emotion and attitude toward LMS, social presence, personal identity, and agency. Questions regarding the dimension of higher-order thinking, were predominantly based on the definition of [62], and were designed to include seven aspects: understanding, analysis, divergent thinking, critical thinking, summarizing ability, problem solving ability, and creativity. Finally, learning satisfaction was measured by students’ overall evaluation of the blended learning course and LMS. Informed by the existing measurement tools, the questionnaire used in the present study comprised 27 items, that were assigned to five scales: technology acceptance (n = 6), emotional experience (n = 5), social belonging (n = 5), higher-order thinking (n = 7), and learning satisfaction (n = 4). The preliminary analysis revealed that the questionnaire had good instrumental reliability and validity. The detailed information about the questionnaire structure, reliability, and validity are shown in Table 1.

### 3.4. Data Collection and Analysis

#### 3.4.1. Data Collection

Data were collected in a cognitive psychology course at a research university in central China. The course was designed using a blended learning model that consisted of two sessions: a lecture session in a conventional classroom and an asynchronous online discussion session on the Xiaoya platform (http://www.ai-augmented.com/, accessed on 4 January 2023), an LMS that supports multi-device access. After 11 weeks of blended learning, we distributed 112 questionnaires to students and collected 110 valid responses. The questionnaire responses that consisted of the same rating for all items, or finished within one minute, were considered invalid and were removed from the analysis. The participants were 110 undergraduate students (40 male students and 70 female students). The average age of the participants was 19.59 years (SD = 1.02, min: 18, max: 23); the group included 86 first-year students, 4 second-year students, 19 third-year students, and 1 fourth-year student. The participants were from three different undergraduate programs: educational technology, digital media technology, and science education, and were taking this course for the first time. In addition, the personal behaviors, including login time and number of posts, were recorded automatically by the LMS and were extracted from its logfile.

#### 3.4.2. Data Analysis

SPSS 25.0 and AMOS 24.0 were used to test the reliability and validity of the questionnaire. Descriptive statistical analyses and correlational analyses were conducted using SPSS 25.0, to explore the associations between the study variables. PROCESS v3.3 macro (Model 6; [63,64]) was used to test the hypothesized serial mediation model. PROCESS provides ordinary least squares regression-based path analysis, using averages of indicators measuring each construct [63]. PROCESS was used because significant associations between variables are not required, bootstrapping reduces type I errors, and it can be applied to small samples [63,65,66].

A direct effect (c’), is the relationship between X and Y controlling for all mediators, and a specific indirect effect, is the relationship between X and Y via a particular mediator or mediators. The value of technology acceptance was entered as the predictor variable (X) and learning satisfaction as the outcome variable (Y). Multiple mediators were then entered twice, in the following order: emotional experience, social belonging, and higher-order thinking; login time and posts. A 5000 sample bootstrapping technique was used to test the indirect effects (the indirect effect was considered to be significant when the bootstrapping 95% CI did not include zero) [63].

## 4. Results

### 4.1. Preliminary Analysis Results

#### 4.1.1. Control for Common Method Bias

Since the questionnaires were self-reports, Harman’s single-factor approach was used to test the common method bias (CMB). Malhotra, Kim [67] suggested that confirmatory factor analysis (CFA) be carried out, to place all test indicators in a “single-factor-model” and test the fitting indicators. The goodness-of-fit indices indicated a poor fit for the single factor model (χ^2^/df = 3.467, CFI = 0.652, TLI = 0.619, RMESA = 0.150), suggesting that biasing from common method bias is unlikely [68]. Hence, there was no serious commonality in the research.

#### 4.1.2. Reliability and Validity

According to Nunnally et al. [69], a Cronbach’s alpha value larger than 0.7 indicates acceptable reliability. As shown in Table 1, the Cronbach’s alpha values of all measurements in the model were greater than 0.8, indicating a good reliability. CFA was used to test the validity of the research model, including convergence validity and discriminative validity.

According to Fornell and Larcker [70], a research model is considered to have good convergent validity when the factor loading of each measurement index is between 0.50 and 0.95, the composite reliability (CR) is greater than 0.60, and the average variance extracted (AVE) is greater than 0.5. As seen in Table 1, the values are consistent with these requirements, thus convergent validity was met.

According to Fornell and Larcker [70], discriminant validity can be proven if the square root of the AVE value of a construct is larger than its correlation coefficients with other constructs. As shown in Table 2, the √AVE values of the constructs ranged from 0.723 to 0.828, greater than the correlation coefficients, indicating an acceptable discriminant validity of the questionnaire as well. Overall, all measures were found to be adequately reliable and valid.

#### 4.1.3. Descriptive Statistics and Correlations

The descriptive statistics and bivariate correlations among variables are shown in Table 3. These results confirm a significant positive correlation between technology acceptance, emotional experience, social belonging, higher-order thinking, and learning satisfaction. Personal behavior variables were significantly positively correlated with technology acceptance (login time and posts). However, learning satisfaction was only significantly correlated with the posts, but not with login time. Therefore, the mediation effect test of login time was not conducted, indicating hypotheses H2 and H4 are not supported.

### 4.2. Mediation Analysis

We tested the mediating role of emotional experience, social belonging, and higher-order thinking, in the relationship between technology acceptance and learning satisfaction, after controlling for gender. As shown in Table 4, technology acceptance significantly predicted emotional experience (B = 0.520, *p* < 0.001), higher-order thinking (B = 0.258, *p* < 0.01), posts (B = 0.266, *p* < 0.01), and learning satisfaction (B = 0.721, *p* < 0.001) (Table 4). Moreover, emotional experience significantly positively predicted social belonging (B = 0.748, *p* < 0.001), and social belonging significantly positively predicted higher-order thinking (B = 0.269, *p* < 0.01). Finally, higher-order thinking significantly positively predicted learning satisfaction (B = 0.463, *p* < 0.001).

The results of the serial mediating effect of technology acceptance and learning satisfaction are shown in Table 5. Interestingly, only higher-order thinking appeared to play a significant and independent mediating role between technology acceptance and learning satisfaction (95% CI = 0.012 to 0.046). Moreover, technology acceptance had a significant impact on learning satisfaction, through the serial mediating paths of “emotional experience–social belonging–higher-order thinking” (95% CI = 0.004 to 0.116).

These results suggest that higher-order thinking plays a mediating role between technology acceptance and learning satisfaction. In addition, emotional experience, social belonging, and higher-order thinking play serial mediating roles between technology acceptance and learning satisfaction. Through the above analysis, a serial mediating model was established, as shown in Figure 2.

## 5. Discussion

In this section, we provide a discussion of the research findings, with an emphasis on theoretical explanation, meaning interpretation, literature comparison, and implications. The limitations of the current study and suggestions for future research are also discussed in this section.

### 5.1. The Role of Login and Post Behaviors in Technology Acceptance and Learning Satisfaction

Neither login time nor post frequency had a significant mediating effect in this study. Consistent with prior studies, technology acceptance has a direct impact on blended learning behavior [71], but blended learning behaviors were found to have little influence on learning satisfaction. The perceived ease of use and usefulness of technology are considered two crucial motivational variables in technology-enhanced environments [16]. Consequently, college students who find the LMS easy to use, and overall beneficial, tend to demonstrate superior learning behaviors [19]. Interestingly, login time was not significantly related to learning satisfaction, perhaps because the increase in login time did not equate with greater learning time, making login behavior itself unsubstantial on learning satisfaction. The limitation of login time as a predictor of satisfaction is also recognized by You [72], who suggests using a list of elaborated time-based indicators, rather than login time alone, to predict students’ sustained endeavors in, and perception of, online learning.

Similar to login time, this study indicated that posting behavior positively correlated with technology acceptance (*r* = 0.333), but its capacity to predict learning satisfaction was negligible (*r* = 0.191). This finding supports previous literature, showing the mere number of posts cannot adequately reflect the quality of discussion [73,74,75]. According to Koszalka, Pavlov [76], students’ participation in online discussion tends to be shallow, and feature “essay-type posting”, with insufficient meaningful peer interaction. Consequently, counting a student’s number of posts fails to measure social dialogue, which is key to students’ blended learning satisfaction [11]. It is therefore not surprising that posting behavior is not a significant mediator between technology acceptance and satisfaction.

### 5.2. The Mediation Effects of Higher-Order Thinking between Technology Acceptance and Learning Satisfaction

One major contribution of the present study, is that we identified higher-order thinking as the only significant mediating variable between technology acceptance and blended learning satisfaction. This finding suggests that skillful use of technological tools can stimulate students’ higher-order thinking, which may greatly improve students’ learning satisfaction. One possible explanation for this, might be that the multiple forms of communication in the technology platform provide better conditions for higher-order thinking [77]. Meanwhile, this finding is consistent with previous studies, suggesting that the development of students’ cognitive level in learning communities created by online discussion forums, is associated with high perceptions of learning gains and learning satisfaction [77,78].

Despite the mediating effect of higher-order thinking, the fact that technology acceptance affects learning satisfaction is not surprising, and is consistent with results from previous TAM model studies. According to Chen et al. [79], individuals are more likely to demonstrate behaviors that they believe will result in positive benefits than those which they do not perceive as having favorable consequences. Hence, students will use technology platforms when they believe their use can benefit learning.

In contrast with previous literature [57,80], this study revealed that social and emotional factors of blended learning, had an insubstantial mediating effect between technology acceptance and learning satisfaction. One possible explanation for this, is that adult learners have good self-learning skills compared to K-12 students [18], suggesting that they can regulate their emotion during the online learning activities and complete the learning task individually. In addition, the influence of online presence and emotional experiences became less significant, since such experiences could be remediated by face-to-face instruction in the blended learning [46].

### 5.3. The Serial Multiple Mediating Effects of Emotional Experience, Social Belonging, and Higher-Order Thinking

The present study showed that college students’ technology acceptance to LMS influences blended learning satisfaction through emotional experience, social belonging, and higher-order thinking. Social cognitive theory emphasizes the close relationship between emotion, society, and cognition [79], and this path revealed that technology platforms can stimulate social learning by creating online communities, cultivating better emotional experiences, and a creating a sense of social belonging, all of which further promote students’ perceived higher-order thinking in blended learning. From the perspective of social interaction, the emotional learning climate [81] and social interaction [82] are important antecedents of beliefs about conducting blended learning. Thus, positive emotional experiences and social belonging encourage and stimulate the exchange of ideas, opinions, information, and knowledge in the organization, that will lead to improved higher-order thinking and better learning satisfaction [83]. Furthermore, it is interesting to note that, neither emotional experience nor social belonging directly relates to learning satisfaction, and both require higher-order thinking to influence learner satisfaction. This is consistent with prior literature, that highlights the importance of cognition level in blended learning environments [5]: while emotional and social factors are important, they tend to influence learning outcomes and experience through higher-order thinking.

### 5.4. Practical Implications

Based on the current research findings, we propose the following implications for college students, teachers, and platform developers, to further improve the blended learning practice in a higher education context. For college students, efforts should be made to create an open and pleasant learning community, that fulfills students’ emotional needs and provides a sense of belonging. Such learning communities are crucial for enhanced cognition during the learning process. For teachers, it is important to design meaningful learning activities to engage students during the blended learning process, so that they can develop their higher-order thinking skills and therefore improve their learning satisfaction. Meaningful activities may include online discussion, group work, critical reflection, and peer assessment. Finally, for platform developers, LMSs should provide students with enough useful and easy-to-use learning functions to improve acceptance among students, as the acceptance of LMS promotes blended learning behaviors, emotion, cognition, and satisfaction.

### 5.5. Limitations

There are several limitations of the current study that should be considered when interpreting the results. First, the survey data were self-reported perceptions based on students’ single semester of online learning. Therefore, the inherent limitations associated with self-report measurement, such as poor objectivity and confirmation bias, must be recognized when interpreting the results. Second, the model was validated using sample data gathered from a single course, at one university in China. Therefore, further studies should include more representative samples, to increase the generalizability of the results. Third, the key constructs of blended learning investigated in this study are not exhaustive, and future research should aim to uncover additional determinants of student blended learning satisfaction.

## 6. Conclusions

Drawing upon social cognitive theories, this study sought to clarify the mechanism underlying the relationship between technology acceptance and blended learning satisfaction, considering the influence of emotional experience, social belonging, higher-order thinking, and behaviors (login time and posts), from the perspective of TRD. We demonstrate that technology acceptance has a significant direct relationship with learning satisfaction. Mediation analysis identified two different significant patterns of mediation: higher-order thinking and serial mediating effects of emotional experience, social belonging, and higher-order thinking. There was no significant mediating effect of learning behaviors on learning satisfaction.

## Figures and Tables

**Figure 1 ijerph-20-04442-f001:**
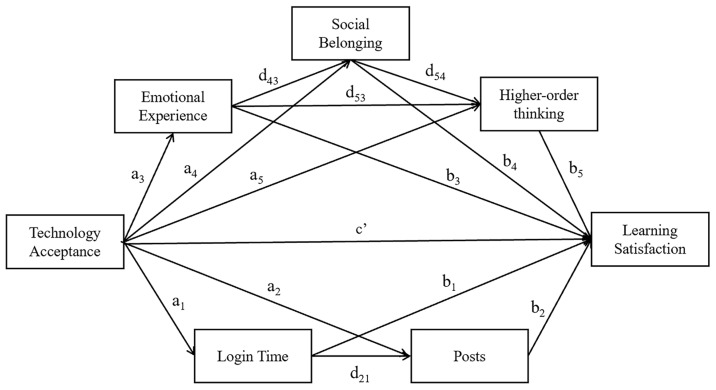
The research model.

**Figure 2 ijerph-20-04442-f002:**
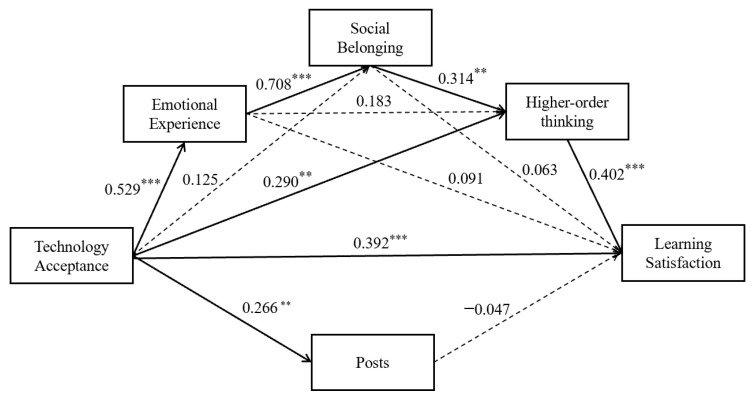
Research model predicting blended learning satisfaction: path coefficients. Note: standardized coefficients are displayed above. Solid arrows: path coefficient is significant, dashed arrows: path coefficient is insignificant. ** *p* < 0.01, *** *p* < 0.001.

**Table 1 ijerph-20-04442-t001:** Reliability and convergent validity analysis.

Construct	Items	Cronbach’s α	Factor Loading	CR	AVE
Technology acceptance	6	0.877	0.627–0.874	0.874	0.539
Emotional experience	5	0.891	0.671–0.816	0.847	0.527
Social belonging	5	0.845	0.701–0.852	0.892	0.623
Higher-order thinking	7	0.809	0.678–0.795	0.817	0.528
Learning satisfaction	4	0.896	0.800–0.882	0.897	0.685

CR, composite reliability; AVE, average variance extracted.

**Table 2 ijerph-20-04442-t002:** Discriminant validity analysis.

Construct	Technology Acceptance	Emotional Experience	Social Belonging	Higher-Order Thinking	Learning Satisfaction
Technology acceptance	0.734				
Emotional experience	0.558	0.726			
Social belonging	0.619	0.884	0.790		
Higher-order thinking	0.661	0.686	0.675	0.723	
Learning satisfaction	0.810	0.617	0.667	0.820	0.828

**Table 3 ijerph-20-04442-t003:** Descriptive statistics and Pearson correlations.

Construct	Mean	SD	1	2	3	4	5	6	7
1 Technology acceptance	3.699	0.746	1						
2 Emotional experience	3.884	0.734	0.509 **	1					
3 Social belonging	3.507	0.776	0.457 **	0.767 **	1				
4 Higher-order thinking	3.898	0.665	0.540 **	0.583 **	0.574 **	1			
5 Learning satisfaction	4.002	0.765	0.689 **	0.570 **	0.545 **	0.705 **	1		
6 Login time	597.05	392.26	0.219 *	0.010	0.015	0.059	0.118	1	
7 Posts	26.409	8.160	0.333 *	0.150	0.063	0.083	0.191 *	0.238 *	1

Note: SD: Standard deviation, * *p* < 0.05, ** *p* < 0.01.

**Table 4 ijerph-20-04442-t004:** Multiple linear regression results among variables.

Model	R²	F (df)	B	Boot SE	t	95% CI	
						Lower	Upper
**Outcome: Emotional experience**	0.266	19.4302_(3,107)_ ***					
Constant			2.045	0.406	6.647 ***	1.303	2.909
Technology acceptance			0.520	0.103	6.224 ***	0.305	0.708
Gender			−0.134	0.120	−1.041	−0.378	0.096
**Outcome: Posts**	0.196	13.065_(3,107)_ ***					
Constant			12.413	4.189	3.466 ***	4.748	21.322
Technology acceptance			0.266	1.025	2.994 **	0.784	4.828
Gender			5.060	1.670	3.370 ***	1.791	8.412
**Outcome: Learning satisfaction**	0.475	48.470_(4,107)_ ***					
Constant			1.441	1.448	5.020 ***	0.594	2.301
Technology acceptance			0.721	0.720	9.368 ***	0.520	0.916
Posts			−0.004	−0.005	−0.603	−0.025	0.013
Gender			0.018	0.022	0.147	−0.207	0.263
**Outcome: Social belonging**	0.609	55.0116_(4,106)_ ***					
Constant			0.251	0.284	0.886	−0.256	0.858
Technology acceptance			0.130	0.081	1.716	−0.039	0.275
Emotional experience			0.748	0.089	9.971 ***	0.576	0.927
Gender			−0.201	0.099	−1.995 *	−0.387	−0.004
**Outcome: Higher-order thinking**	0.451	21.5496_(5,105)_ ***					
Constant			1.301	0.418	4.480 ***	0.553	2.171
Technology acceptance			0.258	0.106	3.301 **	0.047	0.460
Emotional experience			0.166	0.115	1.559	−0.053	0.405
Social belonging			0.269	0.130	2.719 **	0.026	0.528
Gender			0.080	0.102	0.767	−0.118	0.280
**Outcome: Learning satisfaction**	0.644	37.557_(6,104)_ ***					
Constant			0.100	0.359	0.338	−0.595	0.839
Technology acceptance			0.403	0.106	5.253 ***	0.181	0.599
Emotional experience			0.094	0.121	0.940	−0.106	0.367
Social belonging			0.063	0.106	0.654	−0.142	0.276
Higher-order thinking			0.463	0.127	5.091 ***	0.201	0.699
Gender			0.037	0.097	0.383	−0.150	0.223

Note: As recommend by Bolin [61], nonstandardized regression coefficients are reported above. Boot SE, bootstrap standard error. * *p* < 0.05, ** *p* < 0.01, *** *p* < 0.001. CI, confidence intervals.

**Table 5 ijerph-20-04442-t005:** Serial mediating effect of technology acceptance and learning satisfaction.

Path	Effect	Boot SE	95% CI	
			Lower	Upper
Total effect	0.690	0.075	0.244	0.540
Direct effect	0.392	0.105	0.172	0.587
Total indirect effect	0.298	0.104	0.123	0.528
H3: X→M1→Y	−0.013	0.030	−0.073	0.041
H5: X→M2→Y	0.048	0.065	−0.049	0.207
H6: X→M4→Y	0.117	0.062	0.012	0.046
H8: X→M3→Y	0.008	0.016	−0.022	0.046
H7: X→M2→M4→Y	0.039	0.031	−0.009	0.112
H9: X→M2→M3→Y	0.024	0.041	−0.057	0.108
H10: X→M3→M4→Y	0.016	0.015	−0.004	0.054
H11: X→M2→M3→M4→Y	0.047	0.029	0.004	0.116

Note: 5000 bootstrap samples were used. CIs that contain zero are interpreted as nonsignificant; X: technology acceptance, M1: posts, M2: emotional experience, M3: social belonging, M4: higher-order thinking, Y: learning satisfaction.

## Data Availability

The data presented in this study are openly available in Mendeley Data, at https://www.doi.org/10.17632/yr4y8cxw6m.1 (accessed on 5 January 2023).

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
