# Peer review of "Effect of Technology Acceptance on Blended Learning Satisfaction: The Serial Mediation of Emotional Experience, Social Belonging, and Higher-Order Thinking"

_ijerph, 2023, doi:10.3390/ijerph20054442_

Round 1

Reviewer 1 Report

The presented article entitled “Effect of Technology Acceptance on Blended Learning Satisfaction: The Serial Mediation of Emotional Experience, Social Be-3 longing, and Higher-Order Thinking” is interesting. The study explored the relationship between technology acceptance and learning satisfaction in the context of blended learning and examined the influence of related factors (e.g. behavior, emotion, social belonging, higher-order thinking). The topic, in my opinion, is in the scope of interest of the IJERPH readers. The findings of the study may contribute to a better understanding of the design, development, and implementation of blended learning.

There are some places that the authors could clarify to improve the manuscript for possible publication.

First, in the literature review, the authors identified that existing literature had reported the relationships between technology environment, behavior, and learning satisfaction. The authors could clarify how the relationship between learner behaviors and learning satisfaction may differ between and among online learning, technology-enhanced face-to-face learning, and blended learning. And why the research in the context of blended learning is important and needed.

Second, when describing the instruments, more clarity could be included. For example, how many items were included in each subscale, were all the items from existing scales, or were there any changes/adaptions they made? The authors may consider including the questionnaire in Appendix so the readers can also have access to the instruments.

Third, in the discussions, the authors stated how their results supported the findings of previous research. To go beyond that, they could discuss more what new things did we learn from this study compared with current, significant research. In the conclusions, the authors could discuss more how general are their results and findings. 

Reviewer 2 Report

The article presents an interesting topic. The article would gain consistency if:

- consider placing a component with the research objectives and hypotheses together;
- improve the methodology, by identifying the methodological approach and basing it;
- consider relating the various constructs about the description of results (Technology Acceptance, Emotional Experience, Social Belonging,  Higher-Order Thinking and Learning Satisfaction) and not independently;
- add a final part with the conclusions of the article.
